# Starch Films Reinforced with Pistachio Shell Particles: A Sustainable Biocomposite

**DOI:** 10.3390/polym17141907

**Published:** 2025-07-10

**Authors:** Cynthia G. Flores-Hernandez, Alicia Del Real, María de los Ángeles Cornejo-Villegas, Beatriz Millán-Malo, Gerardo A. Fonseca-Hernández, José Luis Rivera-Armenta

**Affiliations:** 1Departamento de Ciencias Básicas, Tecnológico Nacional de México/Instituto Tecnológico de Querétaro, Av. Tecnológico s/n Esq. Gral. Mariano Escobedo, Col. Centro Histórico, Santiago de Querétaro C.P. 76000, Querétaro, Mexico; cynthiagraciela84@gmail.com; 2Centro de Física Aplicada y Tecnología Avanzada, Universidad Nacional Autónoma de México, Boulevard Juriquilla 3001, Santiago de Querétaro C.P. 76230, Querétaro, Mexico; adelreal@fata.unam.mx (A.D.R.);; 3Laboratorio de Procesos de Transformación y Tecnologías Emergentes de Alimentos, Departamento de Ingeniería y Tecnología, FES-Cuautitlán, Universidad Nacional Autónoma de México, Cuautitlán Izcalli C.P. 54714, Estado de México, Mexico; 4Centro de Investigación en Petroquímica, Instituto Tecnológico de Ciudad Madero/Tecnológico Nacional de México, Pról. Bahía de Aldair y Ave. de las Bahías, Parque de la Pequeña y Mediana Industria, Altamira C.P. 89603, Tamaulipas, Mexico

**Keywords:** starch, pistachio shell, biocomposite, thermal analysis, tensile testing

## Abstract

This study investigates the development of corn starch-based biocomposites reinforced with pistachio shell powder, focusing on improving their mechanical and thermal performance. Composite films were prepared by solution casting with pistachio shell contents ranging from 2 wt% to 8 wt% by weight. The materials were characterized using Fourier-transform infrared spectroscopy (FTIR), thermogravimetric analysis (TGA), X-ray diffraction (XRD), and tensile testing. The incorporation of pistachio shell particles led to a progressive improvement in tensile strength and elastic modulus, with the highest values observed in the formulation with 8% reinforcement (SP08). The TGA results indicated a shift in degradation temperatures for the sample with the highest percentage, reflecting a higher thermal stability that is attributed to the interactions between the starch, plasticizer, and cellulosic components of the pistachio shell. The FITR spectra shows very similar structures between starch and pistachio. An XRD analysis shows the alpha-type structure for starch and the cellulose type 1 structure for pistachio. Overall, the results suggest that pistachio shell powder can serve as an effective natural reinforcement, improving the functional properties of starch matrices and promoting the development of environmentally friendly materials derived from agro-industrial waste.

## 1. Introduction

Polysaccharides, including starch, are biodegradable natural polymers composed of monosaccharide units linked by glycosidic bonds. Due to their renewable nature and biocompatibility, polysaccharides are highly valued in materials science as alternatives to conventional plastics [1]. Starch, primarily composed of amylose and amylopectin, is widely utilized for its film-forming abilities and affordability. However, its inherent hydrophilicity and limited mechanical strength can restrict its application in areas where durability and stability are required [2,3]. Many studies have, thus, focused on enhancing starch’s properties through reinforcement with natural fibers, which results in biocomposites with improved mechanical, thermal, and barrier characteristics [1,2,4,5,6].

Pistachio shell, an agro-industrial byproduct, is a lignocellulosic material rich in cellulose, hemicellulose, and lignin. It has shown promise as a natural reinforcement in polymer matrices due to its rigidity, biodegradability, and abundant availability [7,8]. The use of pistachio shell waste has been studied widely in biocomposites for packing applications, and it has good antioxidant and antimicrobial properties [9,10,11,12]. Romero-Ceron et al. studied the effect of modified pistachio shell wastes as reinforcement, reporting that the storage modulus increases when modified pistachio shell was added. Also, they report that there is an interaction between starch and modified pistachio shell, using thermogravimetric analysis and infrared spectroscopy. Also, an increase in crystallinity was found, and they propose an application in the packing industry [13]. Another kind of biopolymer that was improved with the addition of pistachio shell is chitosan. The addition of pistachio shell methanol extract shows significant antimicrobial activity. Also, the FTIR analysis showed that there is good intermolecular interaction between chitosan and pistachio. Furthermore, the thermal stability of chitosan films was improved with the addition of pistachio [14]. A film of chitosan reinforced with pistachio shell powder was reported, and good biocompatibility and antibacterial properties were reported. The mechanical properties were improved with the addition of pistachio shell powder, as well as the swelling properties and thermal stability [15]. By incorporating pistachio shell particles into starch matrices, researchers aim to address starch’s limitations by enhancing the films’ mechanical properties and reducing their water sensitivity, making them viable for biodegradable packaging applications [16,17]. Moreover, cellulose nanocrystals extracted from pistachio shells can form a percolation network within the matrix, which further strengthens and stabilizes the biocomposite structure [18,19,20,21].

This research highlights the significance of valorizing agro-industrial waste by incorporating pistachio shell residues—by-products of food processing—as reinforcing agents in a corn starch matrix. The aim is to develop and characterize biocomposites that not only exhibit suitable structural, thermal, and mechanical properties but also offer a sustainable alternative to conventional synthetic polymers [21,22,23]. By transforming an abundant and underutilized natural residue into a functional reinforcement, this study contributes to the advancement of biodegradable materials and supports global efforts to reduce environmental pollution and dependency on fossil resources. The integration of such biowaste into starch-based systems demonstrates a promising path toward more circular and eco-conscious material design [17,24,25].

## 2. Materials and Methods

### 2.1. Materials

Cornstarch was purchased from Maizena (Unilever Manufacturera, Tultitlan, Mexico). Glycerin was acquired from La Corona industry (100% pure, Parque Industrial Xalostoc, Ecatepec, Mexico), and pistachio shells (*Pistacia vera*) from Wonderful (Los Angeles, CA, USA) were obtained from the waste derived from the production of homemade food.

### 2.2. Obtaining Reinforcer

The pistachio shell powder (PSP) was washed with distilled water and then dried. Afterward, they were crushed using a high-speed multipurpose disintegrator until a fine powder was obtained, which was then passed through a 0.18 mm sieve.

### 2.3. Composite Preparation

To 50 mL of distilled water was added 1 mL of glycerin, 100 and 2 g of starch and pistachio shells particles were added at 2–8% (*w*/*w*). The mixture was heated with magnetic stirring to 90 °C, while covering the beaker with a watch glass to prevent evaporation, and kept at this temperature for 10 min. The mixture was transferred to another heating plate and stirred for 20 min at room temperature. It was then poured into a silicone mold and left to dry at room temperature for 48 h [26]. Table 1 shows the corresponding concentrations and nomenclature used in the synthesis and characterization of composites.

### 2.4. Characterization Methods

Corn starch and pistachio shell powder were characterized by Fourier-transform infrared spectroscopy (FTIR), thermogravimetric analysis (TGA), and X-ray diffraction (XDR) techniques, and the composite films were characterized by TGA, tensile testing, and scanning electron microscopy (SEM).

Measurements were performed using an FTIR spectrometer (Bruker, Tensor 37, Karlsruhe, Germany) with a spectral resolution of 1 cm^−1^ and 32 scans in the wavenumber range of 4000–400 cm^−1^. The sample has a film aspect. The thermogravimetric analyses were carried out with TA Instruments SDT (DSC/TGA, New Castle, DE, USA) model Q600 using a platinum crucible, under a nitrogen atmosphere of 100 mL/min, temperature range 30 to 600 °C, and a heating rate of 10 °C/min with a sample size of 10–15 mg. The X-ray diffraction analyses were performed on a Rigaku model UIV diffractometer with a Cu tube (λCuKα = 1.5406 Å), scintillation detector, and Ni filter. Experiments were carried out in a reflection (Bragg–Brentano) configuration using a fixed slit in a range from 10 to 80° in 2θ with a step size of 0.02° at a sampling rate of 2° per minute for a total of 35 min. It used an aluminum sample holder. Tensile tests were performed on a universal testing machine (Zwick/Roell model Z005, Ulm, Germany) with a load cell of 500 N at a speed of 22 mm/min. The specimens were cut with a microtensile die of the Enclosed ASTM-D-1708 Qualistest brand. Five tests per specimen were performed according to the standard test method for tensile properties for plastic tensile properties by use of microtensile specimens, ASTM D-1708-02a micro-shrinkage specimens. The test conditions were 25 mL/min. The e-module speed was 5 mm/min, humidity 30%, and temperature 22 °C. The morphological and structural characterization of the pistachio shell powder and films was characterized. The scanning electron microscopy (SEM) images were obtained using a microscope (JEOL JSM-6060, Tokyo, Japan) in secondary electron mode at 10 kV with different magnifications. The images were made on the fracture surface and obtained during the mechanical tests and covered with a gold film.

## 3. Results and Discussion

### 3.1. Infrared Spectroscopy (FTIR)

Figure 1 displays the FTIR spectrum of the PSP sample, showing a characteristic polysaccharide region around 1000 cm^−1^, confirming the presence of cellulose and hemicellulose, which are the main components of PSP. This region is similarly observed in the starch film, indicating common C-O-C ring structures and glycosidic bonds within the glucopyranose units [2,27]. The similarities between starch and the PSP spectra arise from their shared glucose disaccharide structures, although starch contains alpha 1-4 glycosidic linkages (noted at 930 cm^−1^), while pistachio shell cellulose exhibits beta 1-4 linkages, typically detected near 1160 cm^−1^ [28].

Starch consists of amylose and amylopectin, whereas PSP includes cellulose and hemicellulose. Notably, both amylopectin and hemicellulose are branched polysaccharides with alpha 1–6 linkages, which appear around 1085 cm^−1^ [29]. Additional distinctive peaks for hemicellulose are found at 2950 cm^−1^ (CH of CH_3_ groups), 1732 cm^−1^ (C=O stretching of methyl ester COOCH_3_), and 1652 cm^−1^ (C-O stretching in carboxylate groups), further differentiating the spectral signatures of PSP, see Table 2 [30].

Given the similarity in polysaccharide structures between starch and PSP and the low pistachio shell content in the composite films, the changes in the FTIR spectra are subtle, which is why they are not shown in the article.

### 3.2. Thermogravimetric Analysis (TGA)

Figure 2 presents the thermogravimetric analysis (TGA) curve (a and c) and its first derivative (DTGA) curve (Figure 2b,d) for corn starch powder. The purpose of this first section (Figure 2a,c) is to identify the thermal behavior and composition of the raw material. This analysis is essential to understanding the baseline degradation profile of the starch, which will later allow for comparison with the biocomposite formulation. The second section (Figure 2b,d) of the study will focus on evaluating the interaction and possible synergy between the starch and the pistachio shell reinforcement. Together, these analyses help explain the relevance of presenting both the TGA and DTGA results in Figure 2, as they provide insights into thermal stability and degradation stages that are critical for assessing the performance of the developed biocomposite. Starch curves reveal a loss of 6.81% between 30 °C to 150 °C, attributed to the evaporation of water content. Lemus et al. 2019, studied the decomposition of amylose and amylopectin from Sigma Aldrich, finding that, at around 300 °C, amylose degrades and, at around 380 °C, amylopectin decomposes. So, we assume, in our analysis, that the second weight loss of 35.5% between 200 °C and 325 °C corresponds to the decomposition of amylose, the third weight loss of 44% between 327 °C and 400 °C is due to the degradation of amylopectin, the fourth weight loss of 2.94% is due to combustion of other organic compound formed in decomposition, and finally, until 600 °C to the decomposition of the amount of carbon from organic sources and other inorganic compounds of 1.15% [37].

Figure 2 shows the thermogravimetric analysis (TGA) and first derivative of the TGA (DTGA) of the PSP. Using both curves, the following is observed in five zones. The first one corresponds to a water loss of 8.95% from 30 °C to 193 °C. The second one occurs from 200 °C to 322 °C, corresponding to a hemicellulose decomposition of 27.05%. The third zone is from 322 °C to 395 °C, attributed to a cellulose decomposition of 41.77%. The fourth zone, from 395 °C to 530 °C, corresponds to the decomposition of organic products of 17.70%, and the final decomposition to obtain ash represents 5.16% [38].

### 3.3. X-Ray Diffraction

Figure 3 presents the X-ray diffraction patterns of (a) corn starch and (b) PSP. Corn starch exhibits crystalline peaks at characteristic 2θ values around 15°, 17°, 18°, and 23°, corresponding to an α-type starch, due to the organization of amylopectin chains into double helices [39,40]. Pistachio shell powder exhibited crystalline peaks at 15°, 17°, 22°, and 35°, corresponding to the indexed crystalline peaks 1–10, 110, 200, and 004, respectively [41], which correspond to cellulose Iβ [42]. In the 2013 article by French and Santiago Cintrón, it is stated that “The three main peaks of the single-stranded triclinic unit cell Iα exhibit Miller indices of (100), (010) and (110) (which are equivalent to the peaks (1–10), (110) and (200) of the cellulose Iβ pattern”, so we conclude that the cellulose present in pistachio shells is of type Iβ.

The degree of crystallinity of the corn starch and PSP samples was determined using JADE v 9.3 software. The peak deconvolution method was used, where the evident peaks were selected from the X-ray diffraction (XRD) pattern. The peaks corresponding to the amorphous region were selected.

For corn starch, three curves were assigned for the amorphous peaks at 11.59°, 19.70°, and 20.70°; the crystalline peaks at 15.12°, 17.20°, 18.18°, 22.96°, 26.54°, and 30.67° were used to perform the corresponding deconvolution. For cellulose, only one amorphous peak, at 20.07°, was used in the amorphous graph [43]. Four crystalline peaks were selected from our spectrum at 14.73°, 16.81°, 22.02°, and 34.55°. Deconvolutions were performed over a range of 10° to 40° in 2θ. A fixed background was used, and the shape and bias parameters were determined using a pseudo-Voigt function, depending on the software. Figure 4 shows the deconvolution curves for (a) starch and (b) pistachio shell powder. The top part of the graph shows the degree of correlation between the deconvolution and the spectrum fit. With this information, the software calculated the crystallinity values: 50.15% for corn starch and 44.04% for pistachio shell powder.

The high degree of crystallinity of pistachio shell cellulose can improve mechanical properties and thermal stability, making it a suitable reinforcement material for biocomposite films [7]. Furthermore, the absence of additional peaks demonstrates the purity of the cellulose, e.g., lignin.

### 3.4. TGA of Composites

Figure 4a shows the DTGA curves, which compare the curves of powdered starch and the starch film. As can be seen, the decomposition temperature (313 °C) of the starch (formed by amylose and amylopectin) is affected by the presence of the plasticizer (glycerol), which was used in the production of the film, resulting in a combination of both peaks present in the powdered starch sample and forming a single peak associated with the decomposition temperature in the curve obtained for the starch film. On the other hand, Figure 4b compares the curves of the pistachio (cellulose and hemicellulose) with the film composed of 8% reinforcement. Changes in decomposition temperatures are also observed in the curves obtained for the composite.

Figure 5 shows the TGA curves of the starch and pistachio shell composite films, which show a slight increase in thermal stability compared to pure starch. Furthermore, Table 3 presents the decomposition temperatures found in the curves, which are attributed to the structures present in the unreinforced composites and films. The onset of decomposition in the composite films shifts to a higher temperature, which is attributed to the presence of cellulose reinforcement present in the pistachio shell, which stabilizes the matrix and delays thermal degradation [44]. Furthermore, a distinctive peak around 293 °C in the composite suggests interactions between the starch and cellulose components, possibly indicating partial crystallinity and stable cross-linking within the composite matrix [45].

### 3.5. Tensile Test for Composites

The mechanical properties (Figure 6) of the developed starch-based films were assessed through tensile strength, elongation at break, and elastic modulus, as summarized in Table 4. The results reveal significant differences in mechanical behavior across formulations, emphasizing the impact of polymer composition and structural reinforcement strategies.

Among the analyzed samples, SP08 exhibited the highest tensile strength (3.39 MPa) and elastic modulus (87.5 MPa) but the lowest elongation at break (27.04%), indicating a rigid and highly interconnected polymer network. This superior mechanical resistance suggests effective load transfer and reinforcement within the matrix, attributed to improved polymer interactions and crosslinking. Similar trends have been reported in starch–protein composites, where higher modulus values correlate with stronger intermolecular forces and reduced ductility [46]. The lower elongation at break in SP08 further supports this, as increased stiffness often results in reduced flexibility [47].

In contrast, SP04 exhibited the lowest tensile strength (1.5 MPa) and elastic modulus (20.3 MPa) but a higher elongation at break (40.2%), indicating a more flexible and less structurally reinforced film. The lower modulus suggests weaker polymer interactions, leading to a more deformable structure. This behavior aligns with observations in starch-based composites, where inadequate phase adhesion and higher porosity reduce mechanical resistance [47,48]. A microstructural analysis supports this behavior, showing a more heterogeneous matrix with dispersed components, which can hinder stress transfer and promote early failure under tension.

The intermediate performance of SP06 (1.64 MPa tensile strength, 22.13 MPa elastic modulus) and SP02 (2.42 MPa tensile strength, 34.38 MPa elastic modulus) suggests a balance between stiffness and ductility. These formulations demonstrate moderate flexibility (elongation: SP06 41.19%, SP02 53.22%), making them viable options for applications requiring a balance between mechanical strength and deformability. The improved elastic modulus in SP02 (34.38 MPa) compared to SP04 suggests better load distribution and chain entanglement, contributing to slightly enhanced mechanical performance. These results align with the findings on biopolymer composites, where starch–polymer interactions influence both stiffness and elongation capacity [46].

Overall, the mechanical performance trends observed in Table 4 highlight the impact of PSP incorporation on the rigidity and flexibility of starch-based films. While SP08 exhibits the highest mechanical resistance, its lower elongation suggests limited flexibility. In contrast, SP04 is more ductile but mechanically weaker, demonstrating the trade-off between stiffness and stretchability in biopolymer composites. Further studies should optimize polymer ratios to enhance both mechanical reinforcement and film flexibility, particularly for applications requiring biodegradable, yet durable, packaging solutions [46,47,48].

### 3.6. Scanning Electron Microscopy (SEM)

Figure 7 displays the SEM micrographs of the pistachio shell powder (Figure 7a), the pure corn starch film (Figure 7b), and the starch–pistachio shell composite (Figure 7c). In Figure 7a, the pistachio shell powder shows a distinctly rough and irregular surface, which may contribute to mechanical interlocking when incorporated into a polymer matrix. Figure 7b, in contrast, presents the morphology of the neat starch film, which appears smooth and homogeneous, indicating a uniform distribution of the polymer without the presence of reinforcements.

Finally, Figure 7c illustrates the surface of the starch–pistachio composite. The microstructure reveals good compatibility and adhesion between the starch matrix and the pistachio particles. No clear signs of phase separation are observed, and the reinforcement appears well embedded in the matrix. These features suggest effective wetting and interfacial bonding, which are important for enhancing the structural integrity and mechanical performance of the biocomposite.

## 4. Conclusions

The incorporation of pistachio shell powder into starch-based matrices led to the development of biocomposite films with improved mechanical and thermal properties. The improvement observed, particularly in the SP08 formulation, is attributed to the reinforcing effect of the cellulosic components present in the PSP, which contributed to higher tensile strength and a higher elastic modulus. Furthermore, the incorporation of PSP led to a progressive improvement in tensile strength and elastic modulus, with the highest values observed in the formulation with 8% reinforcement (SP08). Furthermore, the higher percentage of reinforcement was observed to offer higher tensile strength at the break but lower deformation than the starch-only film. Additionally, the TGA results also indicated a shift in degradation temperatures in the 8% reinforcement formulation, reflecting greater thermal stability attributed to the interactions between the starch, the plasticizer, and the cellulosic components of the pistachio shell. The FTIR spectra of the PSP showed the presence of only cellulose and hemicellulose, and that both pistachio and starch share very similar structures. XRD analysis indicated an alpha-type structure for starch and a type 1 cellulose structure for pistachio. Furthermore, these analyses corroborate the purity of the cellulose components and the absence of lignin or inorganic compounds. Finally, the results suggest that PSP can serve as an effective natural reinforcement, improving the functional properties of starch matrices and promoting the development of environmentally friendly materials derived from agro-industrial waste.

## Figures and Tables

**Figure 1 polymers-17-01907-f001:**
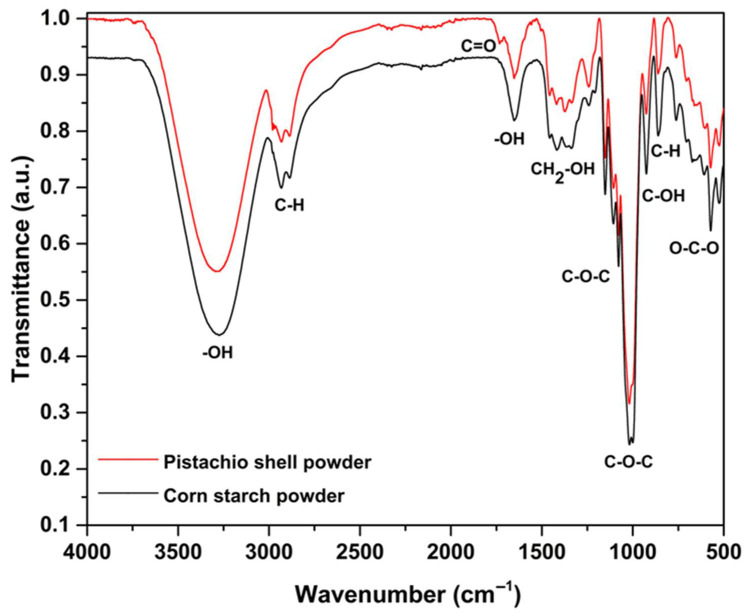
FTIR spectrum of starch and PSP.

**Figure 2 polymers-17-01907-f002:**
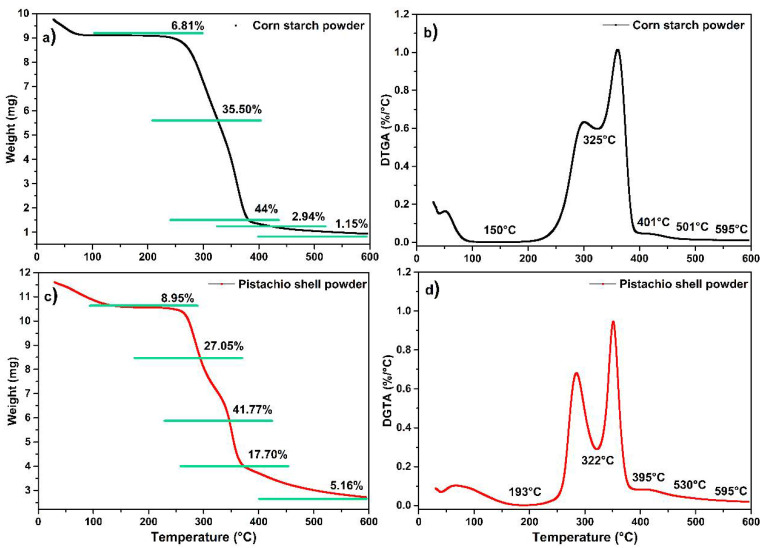
TGA thermograms: (**a**) corn starch powder–TGA, (**b**) corn starch powder–DTGA, (**c**) PSP-TGA, and (**d**) PSP-DTGA.

**Figure 3 polymers-17-01907-f003:**
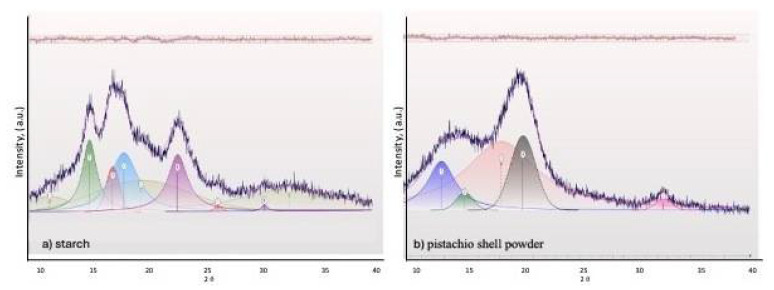
Profile fitting of X-ray diffraction patterns: (**a**) starch and (**b**) PSP.

**Figure 4 polymers-17-01907-f004:**
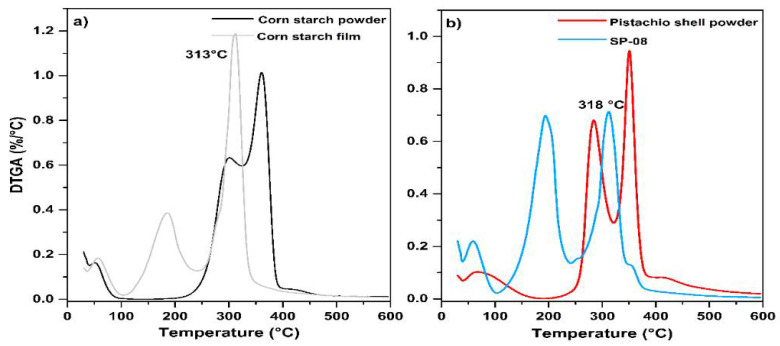
DTGA curves comparative: (**a**) corn starch powder and starch film and (**b**) PSP and AP08 composite film.

**Figure 5 polymers-17-01907-f005:**
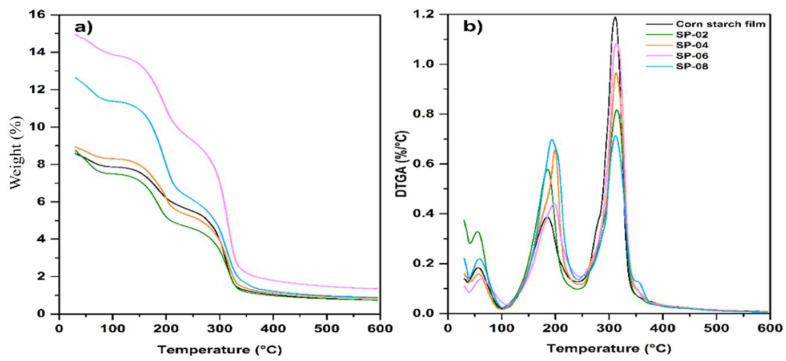
TGA thermograms: (**a**) TGA for corn starch film, starch–pistachio shell composite films, with 2–8 wt % particle content, and (**b**) DTGA for corn starch film, starch–pistachio shell composite films, with 2–8 wt % particle content.

**Figure 6 polymers-17-01907-f006:**
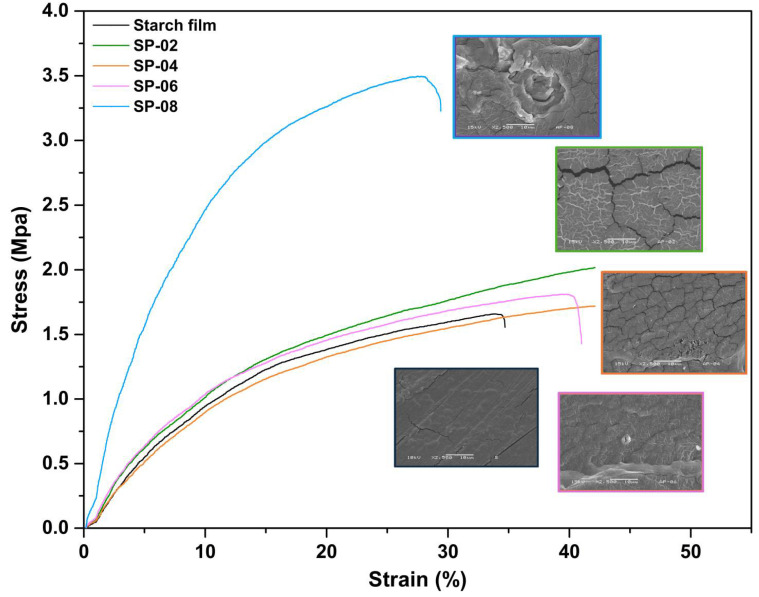
Stress–strain curves for starch pistachio shells composites, with 2–8 wt % of particles.

**Figure 7 polymers-17-01907-f007:**
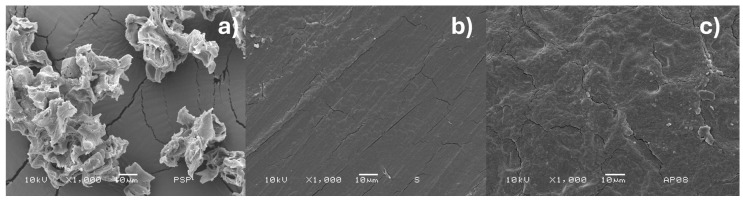
SEM micrographs to 1000×, with 10 micron measuring bar. (**a**) PSP, (**b**) starch film, (**c**) starch–pistachio shells composite (8 wt%).

**Table 1 polymers-17-01907-t001:** Composition and nomenclature of starch–pistachio shell composites.

Percentage of PSP Particles (% wt)	Composites Code
0	Starch
2	SP02
4	SP04
6	SP06
8	SP08

**Table 2 polymers-17-01907-t002:** FTIR signals of starch and PSP.

Starch Powder		Pistachio Shell Powder	
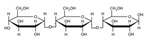	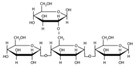	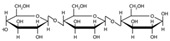	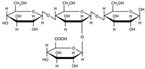
**Amylose**	**amylopectin**	**Cellulose**	**hemicellulose**
3262	-OH stretching of hydroxy group [31,32]	3280	-OH stretching of hydroxy group ^a,b^ and stretching from intermolecular hydrogen bonds [31]
2948	C-H asymmetric stretching of CH_2_-OH [31,32]	2950	C-H asymmetric stretching of CH_2_-OH [31,32]
2930	C-H symmetric stretching of CH_2_-OH [31]	2929	C-H symmetric stretching of CH_2_-OH [31]
2885	C-H symmetric stretching [31]	2886	C-H symmetric stretching [31]
		1733	C=O stretching carboxylic acid [31]
1647	-OH angular bending	1652	-OH bending and carboxylate anions [33]
1434	-CH_2_ deformation of CH_2_-OH [31]	1456	CH_2_ deformation of CH_2_-OH [31]
1415	C-H symmetric stretching of CH_2_-OH [31]	1419	C-H symmetric stretching of CH_2_-OH [31]
		1368	β-glycosidic linkage and asymmetric C-H deformation [34]
1337	C-OH in-plane bending of α-anomer [32]	1334	C-OH in-plane bending of β-anomer [31]
1221	C-O bending of α-anomer [32]	1241	CH_2_ twisting of CH_2_-OH [31]
1149	C-O-C stretching in α-1,4 glycosidic linkage [32]	1152	C-O-C stretching in β-1,4 glycosidic linkage
		1087	C-O deformation in secondary alcohols and aliphatic ethers [35]
1052	C-O-C in α 1,6 glycosidic linkage	1058	C-O-C β-1,3 glycosidic linkage [36]
		1019	C-O and C-C stretching bonds in the C–6 position [36]
1004	C-O-C in-plane bending of α 1,4 glycosidic linkage [32]		
996	C-O stretching in COC α 1,4 glycosidic linkage [32]		
927	C-OH in plane bending [31]	926	C-OH in plane bending [31]
860	C-H deformation pyranose ring [31]	862	C-H deformation in pyranose ring [31]
762	Symmetric breathing of the pyranose ring [31,32]	760	Symmetric breathing of the pyranose ring [31,32]
707	-OH out-of-plane bending [32]	684	-OH out-of-plane bending [32]
586	OCO in-plane bending [32]	584	OCO in plane bending [32]
524	COC in-plane bending in glycosidic linkage and skeletal modes [32]	523	COC in-plane bending in glycosidic linkage, skeletal modes [32]

**Table 3 polymers-17-01907-t003:** Peak decomposition temperatures of composite film components.

	Glycerol 200 °C	Hemicellulose 268 °C	Amylose 306 °C	Amylopectin 364 °C	Cellulose 353 °C
**Starch film**	200 °C		282 °C	318 °C	
SP02	189 °C	264 °C	286 °C	317 °C	
SP04	201 °C	267 °C	288 °C	315 °C	
SP06	201 °C	260 °C	279 °C	316 °C	
SP08	195 °C	253 °C	279 °C	314 °C	350 °C

**Table 4 polymers-17-01907-t004:** Mechanical properties of starch–pistachio shell composites.

Material Code	Elong (%)	Standard Deviation	Elastic Modulus [MPa]	Standard Deviation	Tension (MPa)	Standard Deviation
Starch	33.61	0.54	20.73	7.23	1.66	0.84
SP02	53.22	6.55	34.38	7.63	2.42	0.22
SP04	40.2	4.90	20.3	3.67	1.5	0.28
SP06	41.19	3.57	22.13	2.16	1.64	0.06
SP08	27.04	2.43	87.5	2.78	3.39	0.81

## Data Availability

The original contributions presented in the study are included in the article. Further inquiries can be directed to the corresponding author.

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
