# Peer review of "Starch Films Reinforced with Pistachio Shell Particles: A Sustainable Biocomposite"

_polymers, 2025, doi:10.3390/polym17141907_

Round 1

Reviewer 1 Report

Comments and Suggestions for Authors

This research paper presents a study on the development of biocomposites based on corn starch and pistachio shell powder. The aim of this study is to develop and characterize starch-pistachio shell biocomposites with a focus on their structural, thermal and mechanical properties. Furthermore, the authors evaluate the potential of these biocomposites as sustainable alternatives to petroleum-based materials, promoting biodegradable packaging materials and reducing the environmental impact of synthetic polymers. The new materials were characterized using Fourier transform infrared spectroscopy, thermogravimetric analysis, X-ray diffraction and tensile testing. Overall, the results indicate that pistachio shell powder can serve as an effective natural reinforcement that improves the functional properties of starch matrices and facilitates the development of environmentally friendly materials obtained from agro-industrial waste.

However, the quality of the paper requires some improvement. Below are listed the main comments and questions that, if taken into account, will help improve the quality of the presentation of your work results.

  1. Line 22: «2% to 8% by weight» → «2 wt% to 8 wt%»
  2. Introduction: The presented introduction is uninformative. It is unclear why the focus material is starch, pistachio shells. What are their advantages. In addition, there is no comparison with similar studies. The relevance of the work is not reflected.
  3. Composite preparation, line 19: Incorrect formatting of a literary source.
  4. Figure 1: It is better to place the figure on the next page, since its location in another section of the article confuses the reader.
  5. Figure 1: Incorrect representation of IR spectra of polysaccharides. Cellulose and hemicelluloses cannot be represented as a single material, their structure and properties are significantly different.
  6. Please explain the need for two sections of thermogravimetric analysis.
  7. Section 3.6 is very uninformative. No description. You are stating facts, not scientifically substantiating the results.
  8. Your article title clearly refers to the cellulose product as a reinforcing agent. However, you do not separate the other components from the pistachio shell cellulose. The title needs to be changed to reflect the new emphasis, or additional research needs to be done with purified pistachio shell cellulose.

Author Response

REVIEWER 1

Cellulose-Rich Pistachio Shells as Reinforcement Agents in Starch-Based Films: A Sustainable Biocomposite Strategy

This research paper presents a study on the development of biocomposites based on corn starch and pistachio shell powder. The aim of this study is to develop and characterize starch-pistachio shell biocomposites with a focus on their structural, thermal and mechanical properties. Furthermore, the authors evaluate the potential of these biocomposites as sustainable alternatives to petroleum-based materials, promoting biodegradable packaging materials and reducing the environmental impact of synthetic polymers. The new materials were characterized using Fourier transform infrared spectroscopy, thermogravimetric analysis, X-ray diffraction and tensile testing. Overall, the results indicate that pistachio shell powder can serve as an effective natural reinforcement that improves the functional properties of starch matrices and facilitates the development of environmentally friendly materials obtained from agro-industrial waste.

However, the quality of the paper requires some improvement. Below are listed the main comments and questions that, if taken into account, will help improve the quality of the presentation of your work results.

  1. Line 22: «2% to 8% by weight» → «2 wt% to 8 wt%»

Reply: Thank you for your observation, the necessary correction has already been made. (see page 1, line 23)

  1. Introduction: The presented introduction is uninformative. It is unclear why the focus material is starch, pistachio shells. What are their advantages. In addition, there is no comparison with similar studies. The relevance of the work is not reflected.

Reply: Thank you for your comment. Some paragraphs have been added about pistachios and the relevance of the research. (see page 2, line 8-23 and line 30-39)

  1. Composite preparation, line 19: Incorrect formatting of a literary source.

Reply: Thank you for your comment, the change to the correct format has already been made. (see page 3, line 13).

  1. Figure 1: It is better to place the figure on the next page, since its location in another section of the article confuses the reader.

Reply: Thank you very much for your observation and apology for the confusion, figure 1 has been placed at the end of the FTIR discussion. (see page 4, line 23).

  1. Figure 1: Incorrect representation of IR spectra of polysaccharides. Cellulose and hemicelluloses cannot be represented as a single material; their structure and properties are significantly different.

Reply: Thank you for your comment. Figure 1 shows two infrared spectra: one is from starch, which is composed of amylose and amylopectin, and the second is from pistachio, which is composed of cellulose and hemicellulose. The discussion focuses on the difference between starch and pistachio. Labels have been added to the figure and caption to highlight the differences. (see page 4, line 23).

  1. Please explain the need for two sections of thermogravimetric analysis.

Reply: Thanks for your suggestion, a paragraph has been added at the beginning of the discussion to explain the need for the two TGA sections and why the DTGA charts are required. (see page 6, line 3-12).

  1. Section 3.6 is very uninformative. No description. You are stating facts, not scientifically substantiating the results.

Reply: Thank you for your comment. The discussion of this technique has been expanded and a new paragraph has been added. (see page 11, line 7-19).

  1. Your article title clearly refers to the cellulose product as a reinforcing agent. However, you do not separate the other components from the pistachio shell cellulose. The title needs to be changed to reflect the new emphasis, or additional research needs to be done with purified pistachio shell cellulose.

Reply: Thanks for the suggestion, the title has been changed to “STARCH FILMS REINFORCED WITH PISTACHIO SHELL PARTICLES: A SUSTAINABLE BIOCOMPOSITE”. (see page 1, line 1).

Reviewer 2 Report

Comments and Suggestions for Authors

Ref.:  Ms. No. polymers-3727091

The manuscript entitled “Cellulose-Rich Pistachio Shells as Reinforcement Agents in Starch-Based Films: A Sustainable Biocomposite Strategy”

This study developed corn starch-based biocomposites reinforced with 2–8% pistachio shell powder, showing improved tensile strength and thermal stability. Characterization confirmed strong interactions between starch and pistachio components, highlighting their potential as sustainable, eco-friendly materials from agro-industrial waste. While the study is well-executed, certain aspects need further elaboration to validate its practical relevance fully. I recommend this manuscript for publication in Polymers only after the authors satisfactorily address the following points.

  1. In the introduction, the author should place greater emphasis on the specific significance of this study by highlighting the necessity of cellulose-rich pistachio shells as reinforcement agents compared to other reinforced materials.
  2. In the Experimental section, the author should specify the purity of all chemicals used and ensure consistent reporting of the city and country of origin for each reagent to enhance clarity and maintain uniformity.
  3. In Figure 1, the author should highlight the specific peaks in the FTIR spectrum and clearly label the corresponding regions to enhance clarity and facilitate easier interpretation.
  4. In Figure 2, the author should clarify the meanings of subfigures a, b, c, and d within the figure caption for better understanding. Similarly, for Figure 5.
  5. The author provides the standard deviation for the mechanical properties in Table 4 to assess the reproducibility of the experiments.
  6. The author includes a comparative analysis with other common biocomposite reinforcements, such as rice husk or wood fiber, to contextualize the effectiveness of pistachio shell powder better.
  7. The author should add the scale bar for figures 7a-c.
  8. Reference numbers 13 and 14 are the same.
  9. The author reviews all the references, particularly References 2, 4, 7, 8, and others, to ensure that the missing details, such as page numbers, publication years, and volume numbers, are properly included.

Author Response

REVIEWER 2

The manuscript entitled “Cellulose-Rich Pistachio Shells as Reinforcement Agents in Starch-Based Films: A Sustainable Biocomposite Strategy”

This study developed corn starch-based biocomposites reinforced with 2–8% pistachio shell powder, showing improved tensile strength and thermal stability. Characterization confirmed strong interactions between starch and pistachio components, highlighting their potential as sustainable, eco-friendly materials from agro-industrial waste. While the study is well-executed, certain aspects need further elaboration to validate its practical relevance fully. I recommend this manuscript for publication in Polymers only after the authors satisfactorily address the following points.

  1. In the introduction, the author should place greater emphasis on the specific significance of this study by highlighting the necessity of cellulose-rich pistachio shells as reinforcement agents compared to other reinforced materials.
  2. Reply: Thank you for your comment. Some paragraphs have been added about pistachios and the relevance of the research. (see page 2, line 8-23 and line 30-39).

  1. In the Experimental section, the author should specify the purity of all chemicals used and ensure consistent reporting of the city and country of origin for each reagent to enhance clarity and maintain uniformity.

Reply: Thank you for your comment, the brand, city and country details have been added. (see page 2, line 43-46 and page 3, line 24-35).

  1. In Figure 1, the author should highlight the specific peaks in the FTIR spectrum and clearly label the corresponding regions to enhance clarity and facilitate easier interpretation.

Reply: Thanks for the suggestion, Figure 1 has been modified and the corresponding labels have been added for identification. (see page 4, line 23).

  1. In Figure 2, the author should clarify the meanings of subfigures a, b, c, and d within the figure caption for better understanding. Similarly, for Figure 5.

Reply: Thank you for your suggestion, the captions describing sections a, b, etc., have been added. (see page 7, line 3 and page 9, line 8).

  1. The author provides the standard deviation for the mechanical properties in Table 4 to assess the reproducibility of the experiments.

Reply: Thank you for your suggestion, the standard deviation data has been added to Table 4.  (see page 10, line 22).

  1. The author includes a comparative analysis with other common biocomposite reinforcements, such as rice husk or wood fiber, to contextualize the effectiveness of pistachio shell powder better.

Reply: Thank you for your comment. Some references have been added to research conducted on the use of pistachio shells. (see page 2, line 8-23).

  1. The author should add the scale bar for figures 7a-c.

Reply: Clearer images have been added, indicating the magnification, the 10-micron bar, and the voltage used. These details have also been added to the image caption. (see page 11, line 22).

  1. Reference numbers 13 and 14 are the same.

Reply: Thank you for your feedback, the references have been reviewed. (see page 13, line 40).

  1. The author reviews all the references, particularly References 2, 4, 7, 8, and others, to ensure that the missing details, such as page numbers, publication years, and volume numbers, are properly included.

Reply: The references have been reviewed to complete the missing data. (see page 13-15).

Round 2

Reviewer 1 Report

Comments and Suggestions for Authors

I apologize for the long response to the revised version of the article. Thank you for considering the comments. The article has become more interesting and informative.

Reviewer 2 Report

Comments and Suggestions for Authors

Dear Authors,

Your detailed responses and revisions have significantly improved the clarity and quality of the manuscript.